# Synthetic Peptides Selected by Immunoinformatics as Potential Tools for the Specific Diagnosis of Canine Visceral Leishmaniasis

**DOI:** 10.3390/microorganisms12050906

**Published:** 2024-04-30

**Authors:** Gabriel Moreira, Rodrigo Maia, Nathália Soares, Thais Ostolin, Wendel Coura-Vital, Rodrigo Aguiar-Soares, Jeronimo Ruiz, Daniela Resende, Rory de Brito, Alexandre Reis, Bruno Roatt

**Affiliations:** 1Laboratório de Imunopatologia, Núcleo de Pesquisas em Ciências Biológicas/NUPEB, Universidade Federal de Ouro Preto, Ouro Preto 35400-000, MG, Brazil; gabriel.lucas@aluno.ufop.edu.br (G.M.); rodrigodacostamaia@hotmail.com (R.M.); nathalias.vet@gmail.com (N.S.); thais.ostolin@gmail.com (T.O.); rodrigo.soares@ufop.edu.br (R.A.-S.); rorybrito@gmail.com (R.d.B.); alexreis@ufop.edu.br (A.R.); 2Departamento de Análises Clínicas, Escola de Farmácia, Universidade Federal de Ouro Preto, Ouro Preto 35400-000, MG, Brazil; wendelcoura@ufop.edu.br; 3Programa de Pós-Graduação em Ciências Biológicas, Núcleo de Pesquisas em Ciências Biológicas/NUPEB, Universidade Federal de Ouro Preto, Ouro Preto 35400-000, MG, Brazil; 4Programa de Pós-Graduação em Biotecnologia, Núcleo de Pesquisas em Ciências Biológicas/NUPEB, Universidade Federal de Ouro Preto, Ouro Preto 35400-000, MG, Brazil; 5Grupo de Informática de Biossistemas e Genômica, Programa de Pós-Graduação em Ciências da Saúde, Instituto René Rachou, Fiocruz Minas, Belo Horizonte 30190-002, MG, Brazil; jeronimo@cpqrr.fiocruz.br (J.R.); dani.melo.resende@gmail.com (D.R.); 6Instituto Nacional de Ciência e Tecnologia em Doenças Tropicais, INCT-DT, Salvador 40296-710, BA, Brazil; 7Departamento de Ciências Biológicas, Instituto de Ciências Exatas e Biológicas, Universidade Federal de Ouro Preto, Ouro Preto 35400-000, MG, Brazil

**Keywords:** canine visceral leishmaniasis, serological diagnosis, peptides-based ELISA, immunoinformatic

## Abstract

Diagnosing canine visceral leishmaniasis (CVL) in Brazil faces challenges due to the limitations regarding the sensitivity and specificity of the current diagnostic protocol. Therefore, it is urgent to map new antigens or enhance the existing ones for future diagnostic techniques. Immunoinformatic tools are promising in the identification of new potential epitopes or antigen candidates. In this study, we evaluated peptides selected by epitope prediction for CVL serodiagnosis in ELISA assays. Ten B-cell epitopes were immunogenic in silico, but two peptides (peptides No. 45 and No. 48) showed the best performance in vitro. The selected peptides, both individually and in combination, were highly diagnostically accurate, with sensitivities ranging from 86.4% to 100% and with a specificity of approximately 90%. We observed that the combination of peptides showed better performance when compared to peptide alone, by detecting all asymptomatic dogs, showing lower cross-reactivity in sera from dogs with other canine infections, and did not detect vaccinated animals. Moreover, our data indicate the potential use of immunoinformatic tools associated with ELISA assays for the selection and evaluation of potential new targets, such as peptides, applied to the diagnosis of CVL.

## 1. Introduction

Leishmaniasis belongs to a complex group of diseases caused by a parasite from the *Leishmania* genus, leading to a broad spectrum of clinical presentations [1]. *Leishmania infantum* is one of the primary causes of visceral leishmaniasis (VL), which is fatal in 95% of untreated cases [2]. It is estimated that 50,000–90,000 new VL cases occur annually worldwide, with only 25–45% of these cases being reported to the WHO. In Latin America, VL has been declared endemic in 12 countries, with 63,331 cases being registered between 2001 and 2018 [3]. In Brazil, VL remains a serious public health problem, with 3.466 new human cases reported in 2018, which corresponds to 97% of the total cases in America [3].

Particularly in Brazil, the Leishmaniasis Control and Surveillance Program includes the early diagnosis and treatment of human cases, control of insect vectors, and the identification and culling of seropositive-infected dogs [4]. The management of infected dogs is considered an essential component in the control of VL, given the zoonotic profile of the disease in Brazil [5], where dogs are regarded as urban hosts of the disease, representing an incidence of 5.4 per 1000 dogs-months and a prevalence of 8.1% in endemic areas [6].

Regarding the serodiagnosis of canine visceral leishmaniasis (CVL) in Brazil, the standard protocol recommended by the Brazilian Ministry of Health involves two sequential tests, an initial dual path platform (DPP^®^) test, which is an immunochromatographic test employed in the screening of seropositive dogs, followed by a sequential test, which is an enzyme-linked immunosorbent assay (ELISA) that is considered as the confirmatory test (Brazil. 2014). However, several studies have indicated that this current protocol is inefficient, demonstrating a lower diagnostic accuracy in asymptomatic dogs and those in the initial stages of the infection [7,8]. This represents a serious issue because up to 85% of *L. infantum*-infected dogs may be asymptomatic in endemic areas, acting as a reservoir for the transmission of *L. infantum* between sand flies and humans [9]. Moreover, the tests have yielded false-positive results for dogs infected with other pathogens such as *Ehrlichia canis* and *Babesia canis* [10,11], which could lead to unnecessary euthanasia of noninfected dogs. In addition, it has been reported that the vaccination of dogs might lead to seroconversion, which can directly impact the diagnosis of canine disease.

Despite these limitations, serological methods represent the most practical and flexible tools for epidemiological research and CVL diagnosis. Thus, reinforcing the importance of developing innovative alternatives and identifying new antigens. In this context, immunoinformatics presents itself as a rational tool, given its potential use as a catalyst in the prospecting processes of candidate components of diagnostic tests and vaccines [12,13]. In this study, we propose the use of an immunoinformatic approach to select B-cell epitopes as antigen candidates to be used in CVL serodiagnosis.

## 2. Materials and Methods

### 2.1. In Silico Strategies and Peptide Selection

For a better understanding, a flow chart with an experimental design is shown in Figure 1. The strategy is detailed and explained in the next topics. The peptide sequences were retrieved from the previously established *L. infantum* proteome relational database proposed by Brito et al. 2017 [14]. Using this database, the search simultaneously considered the highest values of 3 B-cell epitope predictor algorithms (AAP12, BCPred12, and BepiPred) [15,16,17,18]; in addition, all intracellular peptides were discarded after considering results obtained from cell location predictor algorithms (WoLF PSORT, Sigcleave, TargetP, and TMHMM) [19,20,21,22]. After that, to elucidate the interactions and metabolic pathways of proteins containing the peptides in the previous step, we used the STRING v.11 algorithms [23,24]; *Leishmania* was established as a reference organism, and text prospecting, experiments, database, co-expression, neighborhood, genetic fusion, and co-occurrence were used as active sources of interaction, with a mean confidence interval of 0.400, and the design of PPI networks was made with the aid of the Cytoscape program, with the addition of information obtained from KEGG [25]. Then, selected peptide sequences were compared against *Leishmania donovani*, *Leishmania braziliensis*, *Leishmania major*, *Trypanosoma cruzi*, *Ehrlichia*, and *Babesia* organisms using the Basic Local Alignment Search Tool (BLASTp) [26]. Furthermore, this same algorithm was used to eliminate proteins that, despite fulfilling the previous criteria, presented a similarity greater than 60% to humans, dogs, and mice, thus decreasing the possibility of obtaining proteins already present in these three organisms.

### 2.2. Peptide Synthesis

Linear peptides from *L. infantum* (15–17 mer) were synthesized with a purity higher than 95% obtained through purification by high-performance liquid chromatography by the Genscript Co., Ltd. (Piscataway, NJ, USA). Once the peptides arrived at the Immunopathology Laboratory, they were stored in an ultra-freezer at −80 °C until use, when they were resuspended (1 mg/mL) in dimethyl sulfoxide (DMSO).

### 2.3. Enzyme-Linked Immunosorbent Assay

To identify the synthetic peptides that offered the highest performance, a peptide-based ELISA was performed, and the conditions were standardized for all 10 peptides. Sera of 10 animals infected with *L. infantum* and 10 healthy animals were used individually for standardization and peptide screening. Flat-bottom polystyrene plates (Nunc MaxiSorp^®^) were sensitized with each peptide, at a concentration of 0.25 µg per well, each well containing 100 µL of carbonate-bicarbonate buffer (pH 9.6), after which they were incubated at 4 °C overnight. After incubation, 4 consecutive washes were carried out with a wash solution composed of PBS (pH 7.2) added with 0.05% Tween 20, to remove antigen excess. Then, using a wash solution added with 5% BSA, possible free sites were blocked. In this step, each well was filled with 100 µL of this blocking solution for 2 h at 37 °C. This was followed by another step of 4 consecutive washes, and, after that, the diluted samples (1:600) were added after dilution in blocking solution (100 µL/well) and incubated again at 37 °C for 1 h. After incubation and 4 consecutive washes, the plates were incubated for 1 h at 37 °C with peroxidase-conjugated sheep anti-dog IgG (Bethyl Laboratories, Inc., Montgomery, TX, USA) diluted 1:16,000 in a wash solution (100 µL/well). The reactions were carried out using 3,3′,5,5′-Tetramethylbenzidine as the substrate for 20 min in the dark, with subsequent interruption of the reaction using 30 µL of 2.5 M H_2_SO_4_, followed by analysis in a spectrophotometer (ELX800 Biotek Instruments, Winooski, VT, USA) at 450 nm. After the screening, two peptides were selected as presenting the best results after the assay with the serum samples. The sequence of the peptides No. 45 (Pep45) and No. 48 (Pep48), described in the current work, is registered at the Instituto Nacional da Propriedade Industrial (Brazil) under patent number BR 1020230118887, deposited on 15 June 2023.

### 2.4. Preparation of L. infantum Soluble Antigenic Extract

The *L. infantum* strain MCAN/BR/2008/OP46 was used for the preparation of the *L. infantum* soluble antigenic extract. Stationary phase promastigotes of *L. braziliensis* were grown at 24 °C in liver infusion tryptose (LIT) medium supplemented with 10% fetal bovine serum (FBS, Sigma-Aldrich, Saint Louis, MO, USA), 100 U/mL penicillin, and 100 μg/mL streptomycin, at pH 7.4. The soluble *Leishmania infantum* antigen (SLiA) was prepared as described previously by [27] Reis et al. (2006) (Reis 2006).

### 2.5. Animal Samples

The study protocol number 083/2007 was approved by the Universidade Federal de Ouro Preto Committees of Ethics in Animal Experimentation. Serum samples from 113 dogs were selected from a serum bank at the Laboratório de Imunopatologia from the Universidade Federal de Ouro Preto, where they were stored at −20 °C. The samples were categorized into distinct groups (Figure 2). Twenty (20) samples from noninfected dogs were included as the control group (CNI; n = 20). This group was composed of five (05) sera from control dogs born in a kennel of the Federal University of Ouro Preto (Minas Gerais, Brazil) and fifteen (15) sera from control dogs from an endemic area in Brazil (Governador Valadares, Minas Gerais, Brazil). The control dogs were characterized by negative parasitological and PCR-restriction fragment length polymorphism (RFLP) results for *L. infantum* in the bone marrow and seronegative results for *Leishmania spp*. using DPP^®^ and BioManguinhos ELISA^®^. The *L. infantum*-infected group of dogs (CVL; n = 37) was divided into three groups based on their clinical status: asymptomatic dogs (AD; n = 18) with no clinical signs of CVL; oligosymptomatic dogs (OD; n = 9) presenting one to three signs; and symptomatic dogs (SD; n = 10) with more than three characteristic clinical signs of VL. The characteristic signs include opaque bristles, a severe loss of weight, onychogryphosis, cutaneous lesions, apathy, and keratoconjunctivitis [27] (Reis 2006). The CVL group was determined based on the serological reactivity of dogs in the BioManguinhos ELISA^®^, DPP^®^, and PCR-RFLP in the bone marrow results. Sera from dogs infected with *E. canis* (n = 15), *B. canis* (n = 9), or *Trypanosoma cruzi* (n = 15) were used for cross-reactivity analyses. Each infection was previously characterized using specific serology (ELISA) and PCR-positive results, and samples were confirmed to be PCR-negative in the bone marrow for *L. infantum.* In addition to the groups described above, the study also used 17 samples from dogs vaccinated with Leish-Tec^®^ (n = 7), a commercial vaccine against CVL available in Brazil, and a potential candidate vaccine, LBSap (n = 10) (Aguiar Soares 2020) [28]. All dogs presented negative serology using DPP^®^ and ELISA and were PCR-negative in the bone marrow for *Leishmania* spp.

### 2.6. Statistical Analysis

The OD cutoff assays were calculated using the receiver operating characteristic curve (ROC curve), by considering the point that yielded the highest combined value of sensitivity and specificity for each antigen. GraphPad Prism software (version 8.0 for Windows) was used to provide the area under the curve (AUC) and the ROC curve. The sensitivity, specificity, negative predictive value (NPV), positive predictive value (PPV), and accuracy were calculated according to Greenhalgh (1997).

## 3. Results

### 3.1. The Database Provides Potential Epitopes for Use in CVL Diagnosis

The database used had 8241 predicted proteins, which when submitted to the B-cell predictor algorithms, returned 47,482 epitopes according to BepiPred, 957,493 according to BCPred12, and 2,361,313 according to AAP12. A search script was then used to find the higher scores for all algorithms, resulting in peptides belonging to five proteins, which were predicted by all three predictive algorithms, and which were secreted/excreted and are not intracellular: LinJ.18.1500, LinJ.32.0970, LinJ.36.2160, LinJ.28.1850, and LinJ. 20.0350. After selection, these proteins have their biological importance analyzed using PPI networks, and all five proteins seem to relate to important cellular functions, especially in metabolic pathways. The peptide sequence, position, prediction scores, and protein ID of each peptide selected are shown in Table 1. The BLASTp analysis was performed to evaluate the identity and similarity between the selected peptides and non-redundant protein sequences from *L*. *donovani*, *L*. *braziliensis*, *L*. *major*, *T*. *cruzi*, *Ehrlichia*, and *Babesia* organisms (Table 2). It is possible to note that for peptides No. 50 and No. 52 to No. 55, no similarity with *Ehrlichia* was found. Furthermore, all peptides showed 100% identity and a low e-value for *L. donovani*. For *L. braziliensis*, the identity values were lower, not reaching 100% for any peptide. The identity for *L. major*, in general, was higher when compared to *L. braziliensis*, and it can still be observed that peptide No. 48 had the highest e-value among all *Leishmania* species and an identity lower than 90%. Regarding sequence similarity with other genera of pathogens, it was observed that peptides No. 46, No. 47, No. 48, No. 50, No. 51, and No. 52 had a high e-value and/or low identity.

### 3.2. All Peptides Were Able to Differentiate Infected and Uninfected Animals during the Standardization Step of the ELISA Reaction

After standardization, and to evaluate the peptides as immunodiagnostic using individual serum samples, ELISA reactions were performed using the 20 previously characterized samples from 10 infected animals and 10 uninfected animals. All peptides showed the ability to distinguish animals infected by *L. infantum*. Using the average of the values obtained in the individual ELISA, the reactivity index showed that each peptide was able to differentiate infected from uninfected animals by 7.94, 7.80, 6.17, 7.51, 6.26, 5.36, 7.19, 3.93, 4.83, and 7.91 times for peptides No. 45–No. 55, respectively (Table 1). By observing this index, it is possible to notice that all peptides significantly separated infected animals from animals not infected by *L. infantum*, with peptide No. 45 being the highest and peptide No. 53 having the smallest difference between optical density values.

### 3.3. The Peptides No. 45, No. 48, and the Combination of Both Showed the Best Capacity to Distinguish L. infantum-Infected Dogs with Different Clinical Forms from the Noninfected Dogs

Considering the results obtained in the ELISA standardization, our selection indicators for the next assays were a peptide with the highest reactivity level and another peptide that showed the lowest OD mean of the negative control. In this sense, the peptides No. 45 (VDPNFQFFHLPVLMF) and No. 48 (FALIRQGFESFPPTPKT) were selected for the next step. We observed that the peptides No. 45 (Pep45) and No. 48 (Pep48), in an isolated form as well as in combination (mix), demonstrated an increased efficiency in differentiating infected dogs from the noninfected, as shown in Figure 3a. Thus, SLiA, No. Pep45, No. Pep48, and mix presented 35/37 (94.6%), 32/37 (86.5%), 37/37 (100%), and 36/37 (97.3%) of the true-positive results, respectively. Therefore, peptide No. 48 and peptide mix were capable of detecting all the asymptomatic dogs (18/18; 100%), oligosymptomatic dogs (9/9; 100% using peptide No. 48, and 8/9; 88.8% using peptide mix), and all symptomatic dogs (10/10) (Figure 3b). The general performance of peptide No. 48 and the peptide mix was superior compared to that of peptide No. 45 and SLiA, demonstrating higher positive (94.8% and 97.3%) and negative predictive values (100% and 95%) and accuracy (both assays presented 96.5%).

### 3.4. The Selected Peptides Present Low Cross-Reactivity with Immunoglobulins from Dogs Infected with Other Canine Pathogens

The ELISA assays showed low reactivity when serum samples from *T. cruzi-*, *E. canis-*, and *B. canis*-infected dogs were tested (Figure 4). No cross-reactivity was observed in *T. cruzi*-infected animals in all tests, in contrast to that with SLiA, which presented 57% positive results for these samples. We observed that the peptide mix showed better results when compared to the results shown by isolated peptides and SLiA. As shown in Figure 4c, the peptide mix presented superior sensitivity (97.2%), specificity (84.7%), NPV (98%), PPV (80%), and AUC (0.9794). Moreover, an accuracy analysis of this approach indicated an excellent performance (89.5%).

### 3.5. Peptides No. 45, No. 48, and Combination Present No Reactivity with Immunoglobulins from the Serum Samples of Vaccinated Dogs

We evaluated the serologic reactivity of serum samples from dogs vaccinated with two vaccines against CVL, Leish-Tec^®^, a commercially available vaccine in Brazil, and LBSap, a potential vaccine candidate that will be commercially available (Figure 5a). We observed that the isolated peptides demonstrated low reactivity for these samples, presenting higher sensitivity, specificity, predictive values, and accuracy compared to that shown by SLiA. However, our data indicate that the mixture of peptides demonstrated a better performance (Figure 5b), high sensitivity (97.2%), high specificity (96.8%), and high accuracy (97.1%), as none of the vaccinated dogs presented with any reaction to the peptide mix, whereas 70.5% of vaccinated dogs were detected positive by SLiA. These results indicated that no reactivity or false-positive diagnosis was observed for the vaccinated dogs when the peptide mix was employed.

## 4. Discussion

Bioinformatics has already shown promise in identifying potential antigens for vaccines or diagnosing infectious parasitic diseases, presenting a reduction in the time needed for this exploration and in the costs, in addition to gaining in diagnostic performance, considering the increase in specificity, sensitivity, and reproducibility indexes [12,29]. In this study, bioinformatics tools were used to select linear epitopes for B-cells based on the combination of three prediction algorithms, BepiPred, AAP12, and BCPred12, based on the assumption that the combination of different algorithms has greater accuracy, especially when dealing with protozoa [30]. Our selected peptides show good score values, including algorithms that are considered more restrictive, which is the case for BepiPred [31,32]. Furthermore, using the PPI networks, it was possible to evidence the biological importance of the proteins that contained the selected peptides, as they can participate in the modulation of cellular activities like virulence, metabolism regulation, or latency behavior in some species of the parasite [33,34,35,36]. The similarity assessment was performed to obtain a previous result of possible cross-reactions between pathogens that can be found in a co-infection situation [10,37]. In this context, a high similarity can be observed with members of the same genus, such as *L. infantum* and *L. donovani.* However, one must pay attention to the similarity in the treatment of the disease when caused by species of the same genus, in addition to the geographic distribution of certain species, such as *L. donovani*, which is not considered prevalent in the New World [38,39]. These observations, added to the lower similarity noted between different genera of parasites, made all the peptides selected by the B-cell prediction algorithms promising for further steps. Thus, all ten selected peptides were used for the standardization steps of the ELISA technique and the assessment of diagnostic capacity.

All 10 peptides showed potential results in the prediction and similarities scores; however, the screening for in vitro evaluation was based on the standardization tests. In this context, we selected peptide No. 45, with a greater difference in reactivity between the mean of infected and uninfected animals that showed the highest reactivity index (7.94), and peptide No. 48, which had the lowest OD mean of the negative control; this threshold can be used to define the lower limits of detection with a reduced cross-reaction between uninfected and infected animals.

Our data indicated that Pep45, Pep48, and a mix of these peptides showed excellent performance, with high sensitivity (86.4%, 100%, and 97.2%, respectively), high specificity (100%, 90%, and 95%, respectively), and elevated accuracy (91.2%, 96.5%, and 96.5%, respectively) in detecting IgG from *L. infantum*-infected dogs presenting different clinical forms. Moreover, these peptides were subjected to a proof-of-concept experiment to test their ability to distinguish leishmaniasis from other infections (*T. cruzi*, *B. canis*, *E. canis*) or vaccinated dogs. Similar studies that employed immunoinformatic tools demonstrated that the peptides, alone or in combination, were reactive to the serum of infected dogs, with accuracy values ranging from 99.6% to 100%; their assay was highly sensitive and specific when compared to the soluble *Leishmania* antigen, which showed low sensitivity and specificity [40]. These peptides, selected using immunoinformatic tools, display superior performance when compared to other studies that have used crude *Leishmania antigen* preparations such as the commercial kit for canine leishmaniasis diagnosis in Brazil, which uses antigens prepared from *Leishmania major*-like promastigotes [41,42,43]. The antigens used in these assays display variability in accuracy parameters owing to antigen preparation and antigenic differences among *Leishmania* species. These considerations were consistent with the present study’s results, in which the statistical parameters of ELISA using the soluble antigen of *L. infantum* (SLiA) displayed inferior performance.

Our results for Pep48 and the peptide mix indicate an efficiency in the detection of most of the infected dogs, including the asymptomatic ones, increasing the odds of detecting infected dogs that might not be detected using other serological tests. The current protocol recommended by the Brazilian Ministry of Health for CVL diagnosis has several limitations. The specificity of the immunochromatographic assay (DPP^®^-CVL) presented specificities varying from 70% to 91.7% [44,45], whereas ELISA (EIE^®^-LVC) presented specificity values varying from 87.5% to 91.76% [44,46].

In this study, we demonstrate that the peptides No. 45, No. 48, and mix presented no reactivity in samples of *T. cruzi*-infected animals, indicating that these peptides may not be shared between *Leishmania* and *T. cruzi* species. In human VL, several studies have shown issues related to cross-reactivity with Chagas disease, owing to the sharing of antigens associated with phylogenetic proximity [47,48]. However, we demonstrated that the mixture of peptides increased the specificity and accuracy compared to that shown by isolated peptides, as confirmed by the higher performance values and lower cross-reactivity in *T. cruzi*, *E. canis*, and *B. canis* pathogens [49], and have previously reported that the association of antigens can lead to better performance because of the use of a single platform for the simultaneous serological measurement of antibodies against different antigens of *L. infantum*. This fact can be associated with the binding of antibodies with higher avidity to the antigens, and the peptide mix could select these antibodies more effectively, favoring a more specific interaction. Despite the lack of concrete evidence of genomic homology between the genus *Ehrlichia* and *Leishmania*, false-positive results are usually found in most conventional serological tests [43,50]. However, some studies do not support serological cross-reactivity among *Leishmania* species, *E. canis*, and *B. canis* (10), based on the phylogenetic differences in such species, since the microorganisms of the genus *Ehrlichia* are intracellular bacteria, whereas *Leishmania* and *Babesia* are protozoa of the orders Kinetoplastida and Piroplasmida, respectively [10,51,52]. Other studies have associated leishmaniasis infection with an increasing co-infection with *E. canis* and *B. canis.* Attipa et al. (2018) reported that dogs with clinical leishmaniasis are 12 times more susceptible to *E. canis* infection when compared to healthy dogs, indicating a suppression of the immune system caused by *Leishmania* infection, which could allow reactivation of a subclinical *E. canis* infection that was present previously or enable the establishment of a new *E. canis* infection, and recommend the simultaneous treatment of both diseases [11,53]. Although, in the present study, each infection was previously characterized using specific serology (ELISA) and PCR-positive results and confirmed to be PCR-negative for *L. infantum*, new mechanisms must be elucidated to understand the cross-reactivity between such species.

The development of vaccines against CVL has been stimulated as an effective and cost-effective strategy for controlling the spread of leishmaniasis in endemic and expanding areas [54]. Although vaccine-induced anti-*Leishmania* antibodies can be detected for several months after immunization, it is essential to evaluate whether a vaccine-induced seroconversion might cause unnecessary culling of noninfected dogs and, consequently, generate problems in surveillance and control programs [55,56]. Both vaccines (Leish-Tec^®^ and LBSap) described in the present study can induce a humoral response [28,57,58]. We observed that peptides No. 45 and No. 48 presented low reactivity with antibodies obtained from dogs immunized with the commercial vaccine, Leish-Tec^®^, and dogs immunized with the LBSap vaccine. The peptide mix showed no reactivity, whereas high reactivity was observed (70.5% of the samples) in vaccinated dogs when SLiA was employed. Our findings indicate that the peptides evaluated had better performance compared to other studies that failed to show low reactivity in samples from vaccinated dogs [59].

In summary, our findings showed that the peptide mix presented a higher performance compared to that shown by SLiA and the isolated peptides, with an excellent capacity for the detection of asymptomatic animals and for discriminating infections with other pathogens or vaccinated dogs, which could improve CVL diagnosis, especially in endemic areas. These data also support the establishment of this approach as the gold standard and open new possibilities for employing these peptides in the construction of chimeric proteins, to improve the accuracy of this test or for the development of a rapid test that could be used as a versatile tool in control programs or in human diagnosis disease.

## Figures and Tables

**Figure 1 microorganisms-12-00906-f001:**
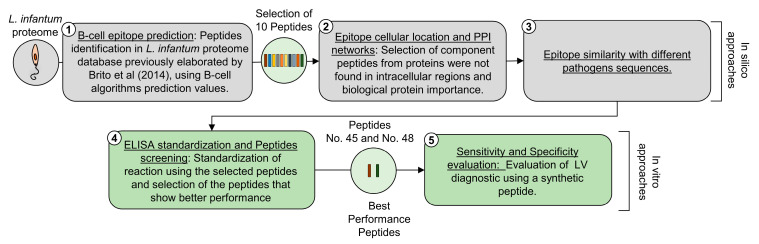
Flow chart with an experimental design for the selection and application of synthetic peptides, based on the epitope prediction approach, in the serological diagnosis of CVL [14].

**Figure 2 microorganisms-12-00906-f002:**
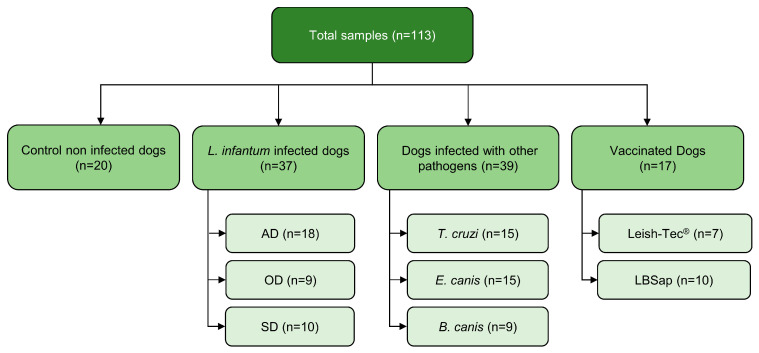
The experimental design employed in the serological test for peptides No. 45, No. 48, and mix. The control noninfected group (CNI), dogs from the endemic area and nonendemic area. The *L. infantum*-infected dogs (CVL) were stratified according to their statuses as asymptomatic dogs (AD), oligosymptomatic dogs (OD), and symptomatic dogs (SD). Vaccinated dogs (LBSap and Leish-Tec^®^) and dogs infected with other pathogens (*Trypanosoma cruzi*, *Ehrlichia canis*, and *Babesia canis*) constitute the other two groups evaluated.

**Figure 3 microorganisms-12-00906-f003:**
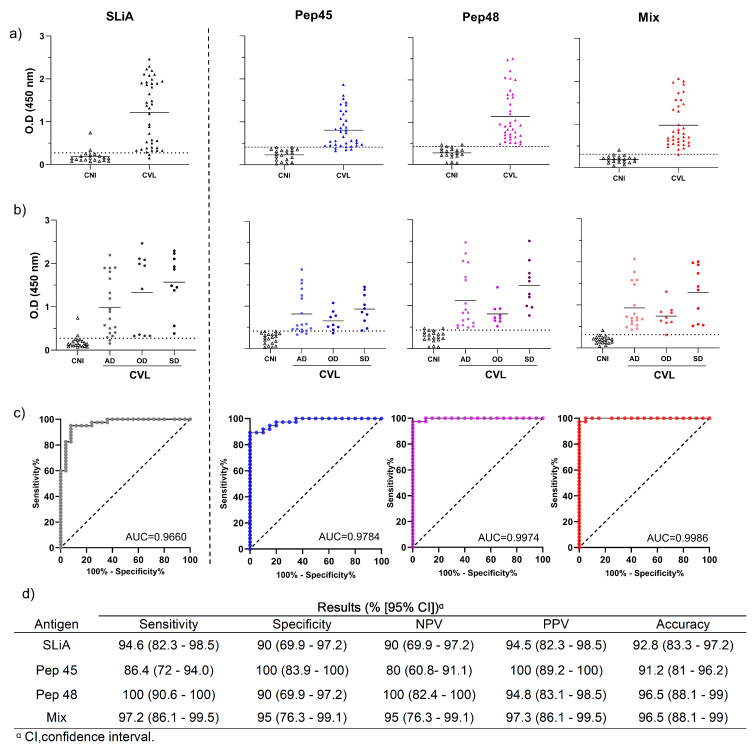
Performance of the peptides No. 45, No. 48, and mix in the discrimination of noninfected and *L. infantum*-infected dogs presenting different clinical forms. (**a**) Distribution of the individual optical density results of the control noninfected dogs (CNI) and *L. infantum*-infected dogs (CVL) tested by the peptides No. 45, No. 48, mix, and soluble *Leishmania infantum* antigen (SLiA). (**b**) Distribution of the individual optical density results of the control noninfected dogs (CNI) and *L. infantum*-infected dogs according to clinical forms. The CVL dogs were stratified according to their clinical statuses as asymptomatic dogs (AD), oligosymptomatic dogs (OD), and symptomatic dogs (SD). The dotted lines within the graphs represent the cut-offs calculated by the ROC curve between the negative and positive results of SLiA (OD = 0.273), Pep45 (OD = 0.41), Pep48 (OD = 0.431), and mix (OD = 0.331). (**c**) The ROC curves were constructed with the results of the control noninfected, and *L. infantum*-infected control serum samples tested by each assay. (**d**) Results of sensitivity, specificity, negative predictive values (NPV), positive predictive values (PPV), and accuracy from the canine visceral leishmaniasis positive and negative animals tested by the peptides No. 45, No. 48, the mix, and SLiA.

**Figure 4 microorganisms-12-00906-f004:**
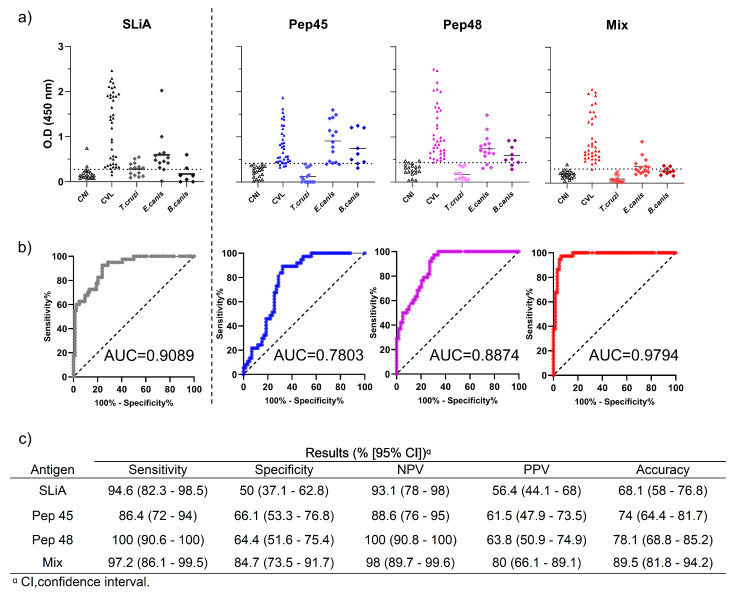
Cross-reactivity of the peptides No. 45, No. 48, and mix with samples from dogs infected with other pathogens of medical and diagnostic importance. (**a**) Distribution of the individual optical density results from serum samples from dogs infected with *Leishmania infantum*, *Trypanosoma cruzi*, *Ehrlichia canis*, and *Babesia canis* using the peptides No. 45, No. 48, mix, and soluble *Leishmania infantum* antigen (SLiA). The dotted lines within the graphs represent the cut-offs calculated by the ROC curve between the negative and positive results of SLiA (OD = 0.273), Pep45 (OD = 0.41), Pep48 (OD = 0.431), and mix (OD = 0.331). (**b**) The ROC curves were constructed with the results of serum samples from control noninfected dogs, *L. infantum*-infected dogs, and dogs infected with other pathogens tested by each assay. (**c**) Results of sensitivity, specificity, negative predictive values (NPV), positive predictive values (PPV), and accuracy from noninfected control dogs, *L.infantum*-infected dogs, and dogs with other canine pathogens tested by the peptides No. 45, No. 48, mix, and SLiA.

**Figure 5 microorganisms-12-00906-f005:**
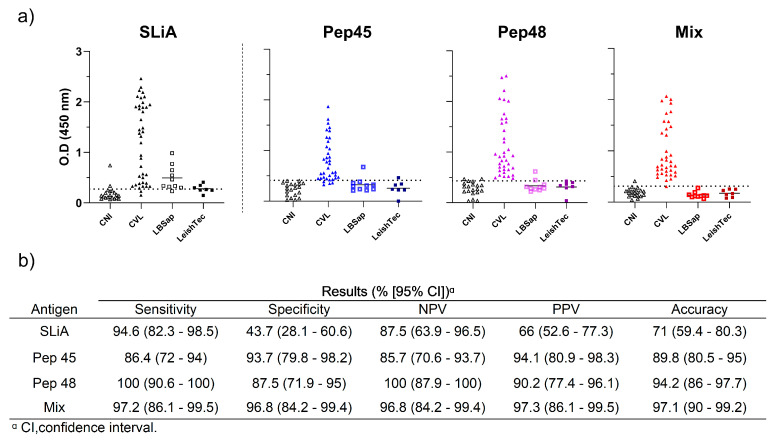
Performance of the peptides No. 45, No. 48, and mix in the discrimination of *L. infantum* infected dogs from vaccinated dogs. (**a**) Distribution of the individual optical density results from serum samples from dogs infected with *L. infantum* and vaccinated dogs using the peptides No. 45, No. 48, mix, and soluble *Leishmania infantum* antigen (SLiA). The dotted lines within the graphs represent the cut-offs calculated by the ROC curve between the negative and positive results of SLiA (OD = 0.273), Pep45 (OD = 0.41), Pep48 (OD = 0.431), and mix (OD = 0.331). (**b**) Results of sensitivity, specificity, negative predictive values (NPV), positive predictive values (PPV), and accuracy from noninfected control dogs, *L. infantum*-infected dogs, and vaccinated dogs tested by the peptides No. 45, No. 48, mix, and SLiA.

**Table 1 microorganisms-12-00906-t001:** B-cell epitopes selected with the highest prediction scores in the databank. The protein identification, peptide sequence, position in protein sequence, prediction scores, and reactivity index are shown.

Protein ID	Peptide ID	Peptide Sequence	Position	Prediction Scores			Reactivity Index
				AAP12	BepiPred	BCPred12	
LinJ.18.1500 (XP_001464963)	45	VDPNFQFFHLPVLMF	692–706	0.9	0.8	0.8	7.94
	46	EGYSSQYYENSWFHRL	763–777	0.9	0.97	0.8	7.80
LinJ.32.0970 (XP_001467777)	47	WAPISEQKGTTYPTTPNGLPV	493–507	1	1.28	1	6.17
LinJ.36.2160 (XP_001469796)	48	FALIRQGFESFPPTPKT	374–390	1	1.67	0.98	7.51
LinJ.28.1850 (XP_001470212)	50	LAVQPAPSTSDAAGA	288–302	0.9	1.47	0.98	6.26
	51	AYQETPESERAELPP	115–129	0.9	1.25	0.98	5.36
	52	LPKGPSVPTLPYQEA	443–457	0.9	1.1	0.99	7.19
LinJ.19.0350 (XP_001464998)	53	SRRPPPLDPEEPEKV	171–185	1	1.74	1	3.93
	54	GLGEEEKEVRQTLRDLR	304–320	1	1	0.96	4.83
	55	CVERITPRVRDRRASYKQS	262–276	1	0.61	1	7.91

**Table 2 microorganisms-12-00906-t002:** Similarity values obtained by BLASTp for each selected B-cell epitope when confronted with the proteome of other species of parasites. Asterisks (*) indicate the absence of a similar sequence; identity values are indicated in percentages (%).

Peptide ID	Similarity											
	*L. donovani*		*L. braziliensis*		*L. major*		*T. cruzi*		*Ehrlichia*		*Babesia*	
	E-Value	Identity	E-Value	Identity	E-Value	Identity	E-Value	Identity	E-Value	Identity	E-Value	Identity
45	8 × 10^−11^	100	1 × 10^−8^	92.86	1 × 10^−9^	93.33	2 × 10^−8^	86.67	23	85.71	14	77.78
46	6 × 10^−12^	100	18	62.5	9 × 10^−9^	87.5	19	66.67	83	85.71	13	100
47	1 × 10^−14^	100	2 × 10^−11^	85	1 × 10^−14^	100	48	61.11	69	76.92	38	71.43
48	2 × 10^−11^	100	2 × 10^−7^	87.5	44	85.71	22	75	11	80	44	100
50	2 × 10^−7^	100	28	69.23	3 × 10^−5^	92.86	61	78.57	*	*	57	90
51	3 × 10^−9^	100	3 × 10^−7^	92.86	3 × 10^−9^	100	16	77.78	33	75	11	53.33
52	7 × 10^−9^	100	2 × 10^−7^	93.33	7 × 10^−9^	100	40	80	*	*	66	64.29
53	2 × 10^−9^	100	1 × 10^−8^	93.33	2 × 10^−9^	100	2 × 10^−5^	80	*	*	16	85.71
54	6 × 10^−11^	100	3 × 10^−9^	94.12	8 × 10^−9^	94.12	2 × 10^−7^	88.24	*	*	39	70
55	9 × 10^−14^	100	2 × 10^−11^	89.47	1 × 10^−12^	94.74	1 × 10^−10^	89.47	*	*	21	77.78

## Data Availability

The raw data supporting the conclusions of this article will be made available by the authors on request.

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
