# Peer review of "Synthetic Peptides Selected by Immunoinformatics as Potential Tools for the Specific Diagnosis of Canine Visceral Leishmaniasis"

_microorganisms, 2024, doi:10.3390/microorganisms12050906_

Round 1

Reviewer 1 Report

Comments and Suggestions for Authors

The paper is well written and contains well described workflow. I have only small remarks:

Sequence of all ten peptides should be given in the paper (not only of peptide No. 45 and No. 48)

The Figure 1 is somehow misleading. Above the first arrow there is a caption “Selection of 10 peptides” and above the last arrow “45 & 48 Peptides”. I suggest switching it to “Peptide No. 45 and No. 48). Otherwise it makes no sense and suggest authors used 10 peptides for screening which resulted in identifying 45 and 48 peptides. I suggest change terms “peptide 45 and 48” to “peptide No. 45 and No. 48” in the whole MS.

I do not insist, but it would be better if the sequences of peptide No. 45 and No. 48 would be given in the abstract. However I understand that Authors apply for patent which  is clearly stated in the body of the text and putting the sequences in the abstract might result in using of the peptides without permission.

Author Response

Reviewer 1: The paper is well written and contains well described workflow. I have only small remarks.

We are thankful to Reviewer #1 for the comments regarding our manuscript and for his critical analysis of our data. As follows, we answer all questions presented.

1) Sequence of all ten peptides should be given in the paper (not only of peptide No. 45 and No. 48);

As requested by the Reviewer #1, we describe the sequence of all tem peptides evaluated in this study.

2) The Figure 1 is somehow misleading. Above the first arrow there is a caption “Selection of 10 peptides” and above the last arrow “45 & 48 Peptides”. I suggest switching it to “Peptide No. 45 and No. 48). Otherwise, it makes no sense and suggest authors used 10 peptides for screening which resulted in identifying 45 and 48 peptides. I suggest change terms “peptide 45 and 48” to “peptide No. 45 and No. 48” in the whole MS.

We agree with the comments of the Reviewer #1. We performed the change of terms, “peptide 45 and 48” to “peptide No. 45 and No. 48” in the whole MS as suggested.

3) I do not insist, but it would be better if the sequences of peptide No. 45 and No. 48 would be given in the abstract. However, I understand that Authors apply for patent which is clearly stated in the body of the text and putting the sequences in the abstract might result in using of the peptides without permission.

 I would like to thanks for the Reviewer #1 understanding about the not inclusion of the sequences of our peptides in the abstract. In fact, we have a chance if we include the sequence in the abstract could result in using of the peptides sequences without permission.

We are extremely thankful to the Reviewer #1, as well as to the Editor for the kindness and attention dedicated to our manuscript. It is truly grateful to hear the outstanding comments that encourage us to keep focusing on the high quality of our future manuscripts.

Reviewer 2 Report

Comments and Suggestions for Authors

The authors indicate the use  of immunoinformatic tools associated with ELISA assays for the selection and evaluation of potential new targets applied to the diagnosis of canine visceral leishmaniasis.

I have one experiment for the authors to justify the immunogenicity of the synthetic peptides

The author should try to isolate the B lymphocytes of the animals and incubate with the synthetic peptides to confirm their ability to induce the B cells by monitoring surface expression of various cell activation markers by flow cytometry.

Another concern regarding the plagiarism check

The authors have to cut down the percent match in their manuscript to below 25%

Comments on the Quality of English Language

Moderate editing of English language required

Author Response

We are extremely thankful to the Reviewer #2, as well as to the Editor for the kindness and attention dedicated to our manuscript. It is truly grateful to hear the outstanding comments that encourage us to keep focusing on the high quality of our future manuscripts.

The point-by-point response follows attached.
